# Diplin: A Disease Risk Prediction Model Based on EfficientNetV2 and Transfer Learning Applied to Nursing Homes

**Feng Zhou** [1] , **Shijing Hu** [1,*], **Xiaoli Wan** [2,*], **Zhihui Lu** [1] **and Jie Wu** [1]

1   School of Computer Science, Fudan University, Shanghai 200438, China; 19110240047@fudan.edu.cn (F.Z.); lzh@fudan.edu.cn (Z.L.); jwu@fudan.edu.cn (J.W.)
2   Information Center, Zhejiang International Business Group, Hangzhou 310000, China
*   Correspondence: sjhu21@m.fudan.edu.cn (S.H.); wanxl@zibchina.com (X.W.)

**Abstract:** In the context of population aging, to reduce the run on public medical resources, nursing homes need to predict the health risks of the elderly periodically. However, there is no professional medical testing equipment in nursing homes. In the current disease risk prediction research, many datasets are collected by professional medical equipment. In addition, the currently researched models cannot be run directly on mobile terminals. In order to predict the health risks of the elderly without relying on professional medical testing equipment in the application scenarios of nursing homes, we use the datasets collected by non-professional medical testing equipment. Based on transfer learning and lightweight neural networks, we propose a disease risk prediction model, Diplin (disease risk prediction model based on lightweight neural network), applied to nursing homes. This model achieved 98% accuracy, 97% precision, 96% recall, 95% specificity, 97% F1 score, and 1.0 AUC (area under ROC curve) value on the validation set. The experimental results show that in the application scenario of nursing homes, the Diplin model can provide practical support for predicting the health risks of the elderly, and this model can be run directly on the tablet.

**Keywords:** transfer learning; image recognition; neural networks; deep learning; health detection

## 1. Introduction

Along with the rapid development of the internet and the massive image data brought by digital cameras, computer vision techniques have been rapidly developed in combination with machine learning techniques. For example, in image classification, the performance of deep learning algorithms approaches or even exceeds that of humans on the ImageNet dataset [1,2]. The innovation of deep learning techniques has enabled AI technologies to be used in various fields. In several fields of the medical industry, medical image data are the more robust data in medical data, and the training of deep learning algorithm models relies on massive data [3]. Therefore, AI-based image detection is widely used in X-ray, CT, and MRI-type image recognition.

Against global aging, population aging has become a regular phenomenon in human society. In its fact sheet on aging and health, the World Health Organization writes: common conditions among older adults include hearing loss, cataracts, refractive errors, back and neck pain, osteoarthritis, chronic obstructive pulmonary disease, diabetes, depression, and dementia. Another hallmark of older age is the development of complex health conditions, often called geriatric syndrome. They are often the result of multiple underlying factors, including weakness, urinary incontinence, falls, confusion, and pressure ulcers. Older adults contribute to their families and communities in many ways, the extent of which depends mainly on one factor: health.

To reduce the run on public medical resources, regular disease risk assessment for the elderly has become an urgent task. However, due to the need for more professional

medical testing equipment in nursing homes, disease risk prediction based on data collected by professional medical testing equipment is unsuitable for the application scenario of nursing homes. In the application scenario of the above evaluation, nursing homes need a prediction method that does not rely on professional medical testing equipment. In this paper, the Diplin model, a disease risk prediction model we propose, does not need to rely on professional medical testing equipment for disease risk prediction. We encapsulate the access to the Diplin model into an API and open access to third-party apps. The application scenario of the Diplin model is shown in Figure 1.

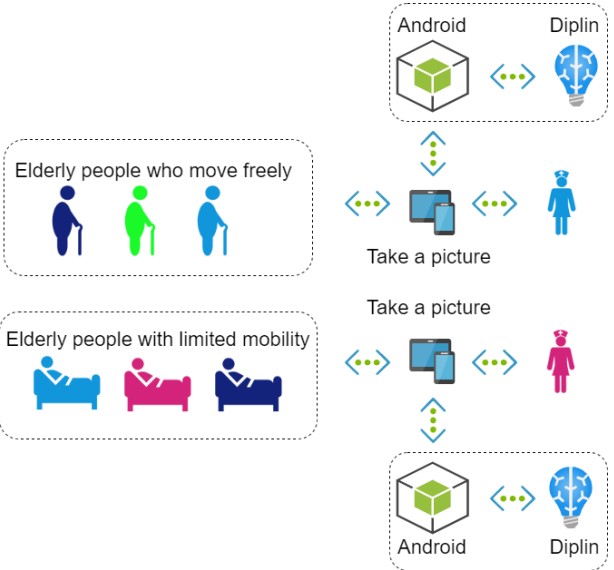

**Figure 1.** Diplin model application scenario.

In Figure 1, we deploy the trained Diplin model directly to the tablet. Nursing home care workers use an app on a tablet to take a photo. After receiving the image data, the Diplin model performs relevant disease risk analyses and returns the disease risk prediction results. Based on the evaluation results, the nursing staff formulates intervention methods for health management.

The research goal of this paper is that the Diplin model can effectively predict the risk of related diseases based on images collected by non-professional medical testing equipment. The Diplin model training uses the datasets "Parkinson's Drawings" and "Oral Cancer (Lips and Tongue) images" publicly available on Kaggle. Both datasets are data collected by non-professional medical equipment. In order to realize the prediction of disease risk in nursing homes, we first use the spiral sub-dataset in "Parkinson's Drawings" and the subject's image data to form a mixed sample data, build and train the sample generation model based on WGAN (Wasserstein GAN), and then build a sample feature preprocessing model. Secondly, we construct and train a sample classification model based on transfer learning and lightweight neural network EfficientNetV2. Finally, we use the wave sub-dataset in "Parkinson's Drawings" and the "Oral Cancer (Lips and Tongue) images" dataset to further verify and test the method of the Diplin model we proposed. The test results' accuracy rate, F1 score, precision rate, and recall rate have all reached more than 90%, and the AUC (area under ROC curve) value has reached 1.0. Ultimately, we realize that using images collected by non-professional medical equipment can predict the risk of related diseases.

The main work of this paper is as follows:

- In the Diplin model, to solve the problem of limited sample number and imbalance, we propose an image sample generation model based on WGAN.
- In the Diplin model, to better protect the image attribute information of the sample itself, we propose an image sample feature preprocessing model.

- In the Diplin model, to reduce the impact of hardware configuration on computing efficiency, we propose an image sample classification model based on transfer learning and the lightweight neural network EfficientNetV2.
- The implementation process of the Diplin model we proposed has good general applicability in the image binary classification task.

This paper has five sections in total. Section 1 is the introduction, which mainly describes the current research background; Section 2 is the related work, which mainly describes the current research status; Section 3 is the model design and implementation, which mainly describes the process of model design and implementation; Section 4 is the result analysis, which mainly describes the analysis of experimental results; and Section 5 is the conclusion, which mainly describes the research results of this paper and the prospect for the future.

## 2. Related Works

There are many studies on disease risk prediction based on artificial intelligence technology. Aditya Khosla et al. proposed an integrated machine learning method for stroke prediction [4]. This method is an automatic feature selection algorithm based on support vector machine (SVM). Jiayu Zhou et al. proposed a multi-task learning formula for predicting disease progression using the cognitive subscale (ADAS-Cog) score in the Alzheimer's disease (AD) assessment scale [5]. This formulation treats the prediction of each stage time point as a task and formulates the prediction of disease progression as a multi-task regression problem. Ankit Agrawal et al. proposed a lung cancer risk prediction model based on support vector machines, artificial neural networks, and random forest technology using lung cancer imaging data from the National Cancer Institute of the United States [6]. Based on this model, they developed an online lung cancer risk prediction system. Alceu Ferraz Costa et al. proposed a feature extraction method for classifying interstitial lung disease in computed tomography (CT) scans based on support vector machines. This method achieved an accuracy rate of 84.36% when performing classification tasks [7].

D. Shiloah Elizabeth et al. proposed an automated method for segmenting lung tissue from chest CT images based on artificial neural networks [8]. This method can be used for preprocessing before the diagnosis of lung diseases to improve the performance of system diagnosis. Md Jamiul Jahid et al. proposed a cancer disease prediction model that can be used in clinical practice using a support vector machine (SVM) as a classifier [9]. Shuo Xiang et al. proposed a multi-source data "two-layer" learning model based on random forests to solve the problem of the loss of block-level data leading to the decline in the prediction accuracy of Alzheimer's disease (AD) [10]. Based on the unified formula, this model handles feature-level and source-level analysis, imputing missing elements.

Roland Assam et al., based on the conditional random field (CRF), used the extracted sample feature vector to capture the latent features of the freeze of gait (FOG) time series of Parkinson's disease patients [11], analyzed the motion time series data of Parkinson's disease patients, and analyzed the patient's freezing of gate state (FOG) for effective prediction. Matthew Seeley et al. used a structured method of integrated learning to compare the accuracy of multiple model combinations for predicting Alzheimer's disease (AD) [12]. Finally, they obtained the characteristic attributes that affect the prediction results. Nida Khateeb et al. used the K-nearest neighbor classifier [13], and by using 14 attributes, the accuracy of heart disease prediction reached nearly 80%. In order to reduce the false alarm rate of pulmonary nodule monitoring, Jiaxing Tan et al. proposed a two-stage deep learning framework based on a deep neural network and a convolutional neural network [14].

Allison M. Rossetto et al. proposed an integrated method based on a convolutional neural network to improve the accuracy of automatic primary diagnosis of lung cancer using deep learning on lung CT [15]. This ensemble method consists of two separate convolutional neural networks. This ensemble method achieved an average accuracy of 85.91%. Based on a convolutional neural network [16], Joongwon Kim et al. designed a

system that can automatically diagnose the risk of lung cancer on chest CT. In order to realize the automatic classification of cystic fibrosis lung disease (CFLD) lesion degree in computed tomography (CT) [17], Xi Jiang et al. proposed a framework based on deep convolutional neural networks and transfer learning. In order to reduce the labeling work in the supervised lung CT image segmentation training [18], Yuan Huang et al. proposed a lung plaque feature extraction method based on a fully convolutional neural network and a generative adversarial network. In order to ensure the high accuracy and performance of fatty liver disease (FLD) prediction [19], Ming Chen et al. proposed a multi-layer random forest (MLRF) model using a medical examination dataset with standardized indicators. This model consists of an input data layer, a processing layer, and an output data layer. Among them, the processing layer comprises multiple random forests (RF).

Hatim Guermah et al. used a dataset containing physiological indicators (such as cell count, red blood cell count, and arterial blood pressure) and dietary attributes to predict the risk of chronic kidney disease based on context ontology and linear SVM [20]. They achieved an accuracy rate of 93.3%. Xuan Chen et al. proposed a weighted loss function to reduce the impact of class imbalance on the prediction results when using pancreatic magnetic resonance images (MRI) for pancreatic cancer risk prediction [21]. Furthermore, based on the ResNet18 model, a classification experiment was carried out, and the experimental results reached an accuracy rate of 91%. Lena Ara et al. used machine learning algorithms to predict peripheral arterial disease [22]. The findings also reduced variability in readouts in vascular laboratories. Amanda H. Gonsalves et al. used naive Bayesian (NB), support vector machine (SVM) [23], and decision tree (DT) to compare the risk prediction of coronary heart disease (CHD). The prediction effect of the model is relatively good.

Anik Saha et al. used a dataset containing physiological indicators (such as blood pressure and albumin) to predict chronic kidney disease (CKD) based on random forest, naive Bayesian [24], and multi-layer perceptron. They obtained an accuracy rate of 97.34%. Md. Golam Sarowar et al. used the "tuberous sclerosis" disease dataset obtained from the National Center for Biotechnology Information (NCBI) based on a hybrid of convolutional neural network (CNN) and particle swarm optimization (PSO) [25]. An optimized CNN algorithm to predict the disease of tuberous sclerosis (TSC) achieved an accuracy rate of 83.47%. Iftikhar Ahmed et al. used the "Myocardial Infarction (MI)" dataset in the UCI machine learning library based on support vector machines [26], multi-layer perceptrons, random forests, additive regression, and ant colony optimization (ACO). Convolutional neural networks (CNN) combine the CNN-ACO algorithm to predict myocardial infarction. The CNN-ACO algorithm achieved an accuracy rate of 95.78%.

When using multi-parameter magnetic resonance imaging (MRI) for prostate cancer (PCa) risk prediction [27], Paulo Lapa et al. found through comparative experiments that the accuracy of classification using semantic learning machine (SLM) is higher than that of CNN XmasNet. Muhammad Mubashir et al. proposed a new method based on a deep convolutional neural network to solve the problem of insufficient classification accuracy due to feature selection and signal analysis using wrist pulses to diagnose lung cancer [28]. This method consists of a 1D fifteen-layer deep convolutional network. This method can identify lung cancer based on the obtained wrist pulse signal and achieved a recognition accuracy of 97.67%. In order to better diagnose lung cancer [29], Yanhao Tan et al. proposed a method for automatically segmenting pulmonary nodules in CT images based on convolutional neural networks. N. Nemati et al. proposed a lightweight classification model based on a deep convolutional neural network and using EEG signals obtained from the CHEG-MIT scalp EEG database for seizure prediction [30], with an accuracy of 99%. Through comparative experiments [31], Rekka Mastouri et al. proved that in CT medical imaging analysis, the detection performance of the fine-tuned model based on the VGG16 model is higher than that of the model formed by transfer learning based on the pretrained VGG16 model.

Chulwoong Choi et al. propose an indexing method based on convolutional neural networks to automatically retrieve lung images from many DICOM medical images generated after PET-CT imaging [32]. This method achieved 70% accuracy. In order to improve the accuracy of detecting COVID-19 through chest X-ray images [33], Jonathan David Freire et al. proposed a new evaluation method based on the Resnet-34 architecture. This method uses data enhancement techniques for image preprocessing, including global histogram equalization and pink mapping. This method achieved an accuracy rate of 97.81%. Based on the Resnet-50 architecture [34], Xiang Yu et al. proposed a model for classifying breast abnormalities using mammographic imaging. In order to improve the accuracy of this model in predicting breast cancer, histogram equalization was used in the data preprocessing stage. Experimental results show that the model achieved an overall accuracy of 95.74%. Yu Lu et al. proposed a lung cancer risk prediction model based on the VGG-16 model and the expanded convolutional pulmonary nodule segmentation network [35]. This model achieved an accuracy rate of 97.1%.

S. I. Lopes et al. used radar-based or infrared thermal imaging technology to perform non-contact monitoring of body temperature and heart rate in nursing homes [36], without connecting physical electrodes, for early detection and prediction of COVID-19 in patients. R. Tsuzuki and others developed an online health chart system based on visualization [37]. This system allows nursing home nurses, medical staff, and older adult family members to view the health trends of the older adult. B. Braga et al. designed a low-cost infrared thermal imaging system for nursing homes for body temperature screening [38]. D. -R. Lu and others have developed an intelligent medical system for nursing homes, which uses the monitoring of service robots to record the current status of the older adult [39]. When the older adult falls, he presses the emergency button on his body to activate the alarm system of the service robot, and the nearby medical staff will receive an emergency call for help.

As shown in Table 1, in the current research on related topics, the accuracy rate of risk prediction for heart disease, lung cancer, chronic kidney disease, pancreatic cancer, tuberous sclerosis, myocardial infarction, epilepsy, and breast cancer has reached 80%. We produced statistics on some studies on the risk prediction of Parkinson's disease, and the statistical results are shown in Table 2.

**Table 1.** Related research statistics.

| Researcher | Disease Name | Basic Algorithm | Accuracy |
| --- | --- | --- | --- |
| Nida Khateeb [13] | heart disease | KNN | 80% |
| Allison M. Rossetto [15] | lung cancer | CNN | 85.91% |
| Hatim Guermah [20] | chronic kidney disease | SVM | 93.3% |
| Xuan Chen [21] | pancreatic cancer | ResNet18 | 91% |
| Anik Saha [24] | chronic kidney disease | Neural Networks | 97.34% |
| Md. Golam Sarowar [25] | tuberous sclerosis | CNN | 83.47% |
| Ifthakhar Ahmed [26] | myocardial infarction | CNN | 95.78% |
| Muhammad Mubashir [28] | lung cancer | CNN | 97.67% |
| N. Nemati [30] | epilepsy | CNN | 99% |
| Xiang Yu [34] | breast cancer | Resnet-50 | 95.74% |

**Table 2.** Comparison of model accuracy on the topic of Parkinson's disease prediction.

| Researcher | Basic Algorithm | Type of Dataset | Accuracy |
| --- | --- | --- | --- |
| Satyabrata Aich [40] | DT | gait data | 81.7% |
| Terry T. Um [41] | CNN | sports data | 86.88% |
| Mehedi Masud [42] | Deep learning | audio data | 96% |
| Anshul Lahoti [43] | RNN | audio data | 83.48% |
| Ours | EfficientNetV2 | image data | 98% |

As shown in Table 2, the accuracy rates of models constructed using decision trees, convolutional neural networks, deep learning, and recurrent neural networks exceed 80%. In this paper, using the dataset collected by non-medical equipment, the prediction accuracy

of the Diplin model proposed by using transfer learning and a lightweight neural network reached 98%.

In the abovementioned previous studies on disease risk prediction, there are many analyses of medical imaging images. However, in the application scenario of a nursing home, the nursing home does not have medical imaging equipment, and it is difficult to ensure the integrity of collecting gait, motion, and audio data. Therefore, we use datasets acquired by non-medical imaging devices in our proposed research work. Our disease risk prediction model based on WGAN, transfer learning, and EfficientNetV2 can run on ordinary tablet computers. In short, research based on datasets collected by professional medical testing equipment are not suitable for the business needs of nursing homes. The Diplin model proposed in this paper does not rely on the datasets collected by professional medical testing equipment, so nursing homes are advised to use the Diplin model.

## 3. Model Design and Implementation

This section mainly describes the design and implementation process of our proposed Diplin model. We first build and train a sample generation model based on WGAN, then build a sample feature preprocessing model; secondly, we build and train a classification model based on transfer learning and lightweight neural network EfficientNetV2; finally, the best model parameters are output according to the evaluation index values such as specificity. Ultimately, we achieve an efficient prediction of associated disease risk through image classification. The software platforms used in this paper are Tensorflow-GPU 2.6.0, Opencv-python 4.7.0.72, Numpy 1.19.5, and Pandas 1.3.5.

### 3.1. Wasserstein GAN

During the training process of GAN (generative adversarial network), there are often problems of non-diverse generated samples and opaque training progress [44]. The root of these problems is that the distance measurement of equivalence optimization needs to be revised, and the distribution generated by the generator overlaps with the accurate distribution. To avoid such problems, we propose the Diplin model, which uses the Wasserstein distance proposed by Martin Arjovsky et al. to measure the distance between the accurate and generated sample distribution [45]. The specific formula expression of Wasserstein distance is shown in Formula (1) [45]:

$$W\big(\mathbb{P}_\gamma, \mathbb{P}_g\big) = \inf_{\gamma \in \Pi(\mathbb{P}_\gamma, \mathbb{P}_\mathcal{G})} \mathbb{E}_{(\mathcal{X},\mathcal{Y})\sim\gamma}[\| \mathcal{X} - \mathcal{Y} \|] \tag{1}$$

In Formula (1) [45], $\Pi\big(\mathbb{P}_\gamma, \mathbb{P}_g\big)$ represents the set of all possible combination distributions of $\mathbb{P}_\gamma$ and $\mathbb{P}_g$ combination; $\gamma$ represents each possible combination distribution; $(\mathcal{X}, \mathcal{Y}) \sim \gamma$ represents a real sample $\mathcal{X}$ obtained by sampling and a generated sample $\mathcal{Y}$; $\| \mathcal{X} - \mathcal{Y} \|$ represents the distance between the accurate sample and the generated sample; and $\mathbb{E}_{(\mathcal{X},\mathcal{Y})\sim\gamma}[\| \mathcal{X} - \mathcal{Y} \|]$ represents the expected value of the distance of the sample under the combined distribution. The Wasserstein distance is the lower bound of this expected value in all possible combined distributions. However, it is not easy to directly solve the Wasserstein distance in GAN. However, the theory of Kantorovich–Rubinstein duality can convert the problem into a dual problem expressed by Formula (2) [45]:

$$W(\mathbb{P}_\gamma, \mathbb{P}_\theta) = \sup_{\|f\|_{L\leq 1}} \mathbb{E}_{\mathcal{X}\sim\mathbb{P}_\gamma}[f(x)] - \mathbb{E}_{\mathcal{X}\sim\mathbb{P}_\theta}[f(x)] \tag{2}$$

In Formula (2) [45], $L \leq 1$ means $f$ is a 1-Lipschitz function. $\mathbb{E}_{\mathcal{X}\sim\mathbb{P}_\gamma}[f(x)] - \mathbb{E}_{\mathcal{X}\sim\mathbb{P}_\theta}[f(x)]$ represents the upper bound value that satisfies the 1-Lipschitz function limit value. Lipschitz represents the maximum magnitude of local variation of a continuous function. The calculation formula for parameter data optimization using the neural network method is shown in Formula (3) [45]:

$$\max_{\mathcal{W}\in\mathcal{W}} \mathbb{E}_{\mathcal{X}\sim\mathbb{P}_\gamma}[f_\mathcal{W}(x)] - \mathbb{E}_{\mathcal{Z}\sim p(\mathcal{Z})}[f_\mathcal{W}(g_\theta(\mathcal{Z}))] \tag{3}$$

In Formula (3) [45], $\mathcal{W} \in \mathcal{W}$ represents an extreme assumption. This assumption represents the assumption when proving the consistency of the estimator. The improvement of WGAN to GAN mainly has the following points [45]:

- The LOSS of the generator network and discriminator network no longer takes LOG.
- Since the discriminator network needs to fit the Wasserstein distance, which is a regression problem, not a classification problem, the last layer of the discriminator network removes the sigmoid.
- Since LOSS is volatile when using momentum-based optimization algorithms (including momentum and Adam), the optimization algorithm recommended by WGAN is MSProp or SGD.
- After each discriminator network parameter is updated, this parameter is truncated so that the absolute value of this parameter does not exceed a fixed value, that is, the function of the discriminator network is a Lipschitz function, and the derivative of this function is less than a particular fixed value.

### 3.2. Transfer Learning

Jason Yosinski et al. proposed the possibility of the transfer of deep neural networks and demonstrated through experiments that the first three layers of deep neural networks are standard features [46]. Migrating the parameters of a trained model to a new training task can save training time—overhead and other resource overhead. In image classification [47], the model weights trained on the ImageNet dataset are generally transferred to new classification tasks. Then, the shallow convolutional and pooling layers are used to construct the sample feature extraction model [48]. Finally, a new classification task model is constructed.

Due to the limited number of image samples in the application scenarios of nursing homes, the Diplin model we proposed uses deep transfer learning. During the model's training, because we are training a binary classifier, the loss function we use is binary_crossentropy. The formula of this binary cross entropy is shown in Formula (4):

$$Loss = -\frac{1}{N} \sum_{i=1}^{N} y_i \cdot \log(p(y_i)) + (1 - y_i) \cdot \log(1 - p(y_i)) \tag{4}$$

In Formula (4), the label value of $y$ is 0 or 1, and $p(y)$ is the probability that the output belongs to the label $y$. If the $y$ label value is 1, if the predicted value is close to 1, then the value of the loss function also tends to be 0. If the predicted value is close to 0, then the value of the loss function will be more considerable.

The similarity of features between the target sample data and the source sample data mainly determines the effect of transfer learning [49]. There will be a certain probability of negative transfer in the training tasks with significant differences between the two sample data types. In this paper, our criterion for judging negative transfer is to compare the performance of the model trained with transfer learning and the model trained without transfer learning. In the case of performance degradation, we fine-tune the model and process it further.

### 3.3. EfficientNetV2

The EfficientNetV1 model surpasses the previous network regarding training effect, model parameter quantity, and training speed [50]. EfficientNet-B7 has a parameter size of 66M but achieved an accuracy rate of 84.3% on ImageNet. However, when there is considerable image input, EfficientNetV1 will take up more video memory, significantly reducing the training speed. In addition, since EfficientNetV1 uses depth-separable convolution, more intermediate variables need to be saved during training than ordinary convolution, which increases the time overhead of reading and writing data and reduces the training speed. In order to solve the problem of decreased training speed caused by video memory usage and intermediate variable storage, Google published a lightweight neural network model EfficientNetV2 on CVPR in April 2021 [51]. This model has been

dramatically improved in terms of training speed and parameter efficiency through the training of perceptual neural structure search (training-aware NAS) and scaling. In this paper, the Diplin model we proposed uses the network structure of EfficientNetV2-S, as shown in Table 3.

**Table 3.** EfficientNetV2-S architecture [51].

| Stage | Operator | Stride | No. Of Channels | No. of Layers |
|---|---|---|---|---|
| 0 | Conv3*3 | 2 | 24 | 1 |
| 1 | Fused-MBConv1, k3*3 | 1 | 24 | 2 |
| 2 | Fused-MBConv4, k3*3 | 2 | 48 | 4 |
| 3 | Fused-MBConv4, k3*3 | 2 | 64 | 4 |
| 4 | MBConv4, k3*3, SE0.25 | 2 | 128 | 6 |
| 5 | MBConv6, k3*3, SE0.25 | 1 | 160 | 9 |
| 6 | MBConv6, k3*3, SE0.25 | 2 | 256 | 15 |
| 7 | Conv1*1 and Pooling and FC | - | 1280 | 1 |

In Table 3 [51], k represents the size of the convolution kernel used; Fused-MBConv represents the Fused-MBConv network; MBConv represents the MBConv network; SE (squeeze-and-excitation) represents the SE attention network; the step size of 2 means that the output width and height of the current stage are half of the input width and height. The Fused-MBConv network and the MBConv network are shown in Figure 2.

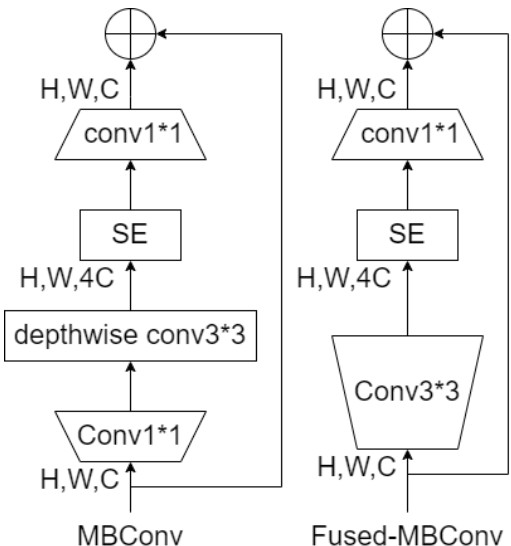

**Figure 2.** Structure of MBConv and Fused-MBConv [51].

As shown in Figure 2 [51], MBConv consists of Conv1*1 dimension-up convolution, depthwise conv3*3 depth convolution, SE (squeeze-and-excitation) attention module, and Conv1*1 dimensionality reduction convolution. The difference between Fused-MBConv and MBConv is that Conv3*3 ordinary convolution is used in Fused-MBConv to replace depthwise conv3*3 depth convolution and Conv1*1 up-dimensional convolution.

*3.4. Diplin Model*

The method architecture of our Diplin model based on WGAN, transfer learning, and EfficientNetV2 is shown in Figure 3.

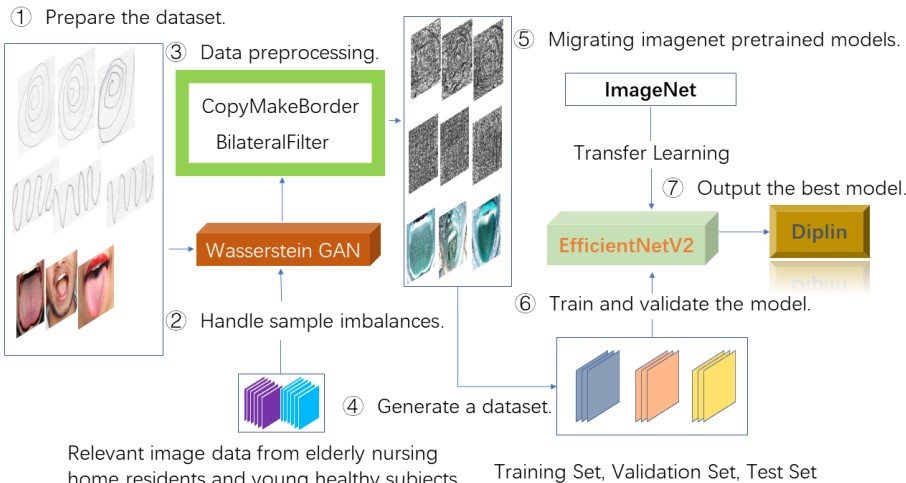

**Figure 3.** Diplin model method architecture.

The detailed process of the Diplin model design process in Figure 3 is as follows:

1.  Prepare the image data of Parkinson's disease patients and healthy people drawing spirals and waves; prepare the image data of lips and tongues of oral cancer patients and healthy people.
2.  Build and train a sample image generation model based on Wasserstein GAN using relevant image data of older adults in nursing homes and young, healthy subjects.
3.  Build a sample image feature preprocessing model, including setting the boundary of the image, data augmentation, and using filtering algorithms to effectively protect the spatial and color information of the edge information in the sample image.
4.  Generate the training dataset, validation dataset, and test dataset.
5.  Load the EfficientNetV2 model trained on ImageNet.
6.  The classification model's dataset, training, and parameter optimization are used based on transfer learning and the EfficientNetV2 model.
7.  After iterative training, the best model is output.

### 3.4.1. Sample Data

In this paper, we use the datasets "Parkinson's Drawings" and "Oral Cancer (Lips and Tongue) images" publicly available on Kaggle. "Parkinson's Drawings" are the image data of spirals and waves drawn by patients with Parkinson's disease and healthy people [52]. "Oral Cancer (Lips and Tongue) images" are the image data of cancerous and non-cancerous lips and tongues collected at various ENT hospitals in Ahmedabad [53]. The details of the data distribution of the "Parkinson's Drawings" dataset are shown in Table 4.

**Table 4.** "Parkinson's Drawings" dataset data distribution [52].

| Category | Category | Category | Quantity |
| --- | --- | --- | --- |
| spiral | testing | healthy | 15 |
| spiral | testing | Parkinson's | 15 |
| spiral | training | Healthy | 36 |
| spiral | training | Parkinson's | 36 |
| wave | testing | healthy | 15 |
| wave | testing | Parkinson's | 15 |
| wave | training | Healthy | 36 |
| wave | training | Parkinson's | 36 |

In Table 4 [52], "Parkinson's Drawings" include Parkinson's disease patients and healthy people drawing spiral and wave class II image data. The spiral and wave folders include two folders: testing, and training, respectively. The testing and training folders

include healthy and Parkinson's folders, respectively. There are 15 sample images, each for healthy and Parkinson's in the testing folder. There are 36 sample images for healthy and Parkinson's in each training folder. The "Oral Cancer (Lips and Tongue) images" dataset includes two folders: cancer and non-cancer. The cancer folder includes 87 sample images. Forty-four sample images are included in the non-cancer folder [53].

### 3.4.2. Sample Image Generation

In sample image generation and implementation, we use five sets of CONV2D and BatchNorm layers to form the backbone network, convolution, and normalization to process shortcuts and deconvolution (Conv2DTranspose) to realize the downsampling of the discriminant network, generate network upsampling. The input of the discrimination network uses accurate sample image data, the noise when generating the sample data, and the mixed data of real and fake data. The input of the generation network uses random noise. Considering that the data in the tfrecord format is convenient to save and transfer, and the reading speed is fast, we use the tfrecord format to save the data in this paper. The main steps are as follows:

- Define upsampling and downsampling functions.
- Define the ResBlock module.
- Build a generated network and a discriminator network [54].
- Define the discriminative network input.
- Mix generated data and accurate sample data.
- Define the discriminative network loss function.
- Define the generator network input.
- Define the generative network loss function.
- Define the training network loop body, and train the generative and discriminative networks.

### 3.4.3. Sample Feature Preprocessing

In this paper, in sample feature preprocessing, we use a filtering algorithm to protect the spatial, noise, and color information of the edge information in the sample image. Considering that the size limit of the convolution operator will cause the boundary of the sample image to be lost, we appropriately expand the boundary of the sample image. The main steps are as follows:

- Sample image border extension.
- Sample image for bilateral filtering.
- Based on the comparison of mean_squared_error image similarity, duplicate sample data are eliminated.
- Sample image normalization.
- The training dataset is generated using a 60% training set, 20% verification set, and 20% test set.

### 3.4.4. Classification Model Construction and Training

This paper uses TensorFlow to build and train classification models based on transfer learning and EfficientNetV2. We first migrate the weights of the EfficientNetV2 model pretrained using the ImageNet dataset to the classification model, then build the classification model network and compile the classification model. Finally, we load the sample image data to train and verify the classification model. The main steps are as follows:

- Migrate pretrained model weights.
- Freeze the base network.
- The backbone model network is constructed using a global average pooling layer, a fully connected layer, and a dropout layer.
- Compile and train the defined model network.
- Unfreeze some layers of the base network.
- Compile and train unfrozen partial layers and define model networks.

In this section, we describe the sample data and ImageNet data used in this paper, the implementation of sample image generation based on WGAN, the implementation of sample feature preprocessing, and the classification model based on transfer learning and EfficientNetV2 implementation process.

3.4.5. Model Algorithm

The method of Diplin model training is shown in Algorithm 1.

---

**Algorithm 1:** Diplin model algorithm.

---

**Input:** $\{s_1, s_2, \ldots, s_n\}$, $\{l_1, l_2, \ldots, l_n\}$, $\{N_1, N_2, \ldots, N_n\}$, $\{Y_1, Y_2, \ldots, Y_n\}$
**Output:** Diplin best model
*Tra*: Training sample data.
*Val*: Verify sample data.
*Test*: Test sample data.

1    Combine $\{s_1, s_2, \ldots, s_n\}$, $\{l_1, l_2, \ldots, l_n\}$, $\{N_1, N_2, \ldots, N_n\}$ and $\{Y_1, Y_2, \ldots, Y_n\}$ to form mixed sample data.
2    Build and train a sample image generation model based on Wasserstein GAN.
3    Build a sample image feature preprocessing model.
4    Sample image data preprocessing.
5    Divide the sample dataset into three parts: one part for training, one part for validation, and one for testing. Obtain training data *Tra*, verification data *Val*, and test data *Test* [55].
6    Load the weights of the EfficientNetV2 model trained on ImageNet.
7    For epochs do
8      Train a classification model.
9      By observing the evaluation index value, parameter optimization is carried out [55].
10      Evaluate the model using specificity, AUC value, etc.
11      If Accuracy $\geq 0.98$ and Precision $\geq 0.97$ and Recall $\geq 0.96$ and Specificity $\geq 0.95$ and F1-score $\geq 0.97$ and AUC $\geq 0.98$:
12        Output Diplin
13        Complete training.

---

In the above algorithm, $\{s_1, s_2, \ldots, s_n\}$ represent the image data of Parkinson's disease patients and healthy people drawn by Kaggle to draw spirals and waves. $\{l_1, l_2, \ldots, l_n\}$ represent the image data of lips and tongues of oral cancer patients and healthy people published by Kaggle. $\{N_1, N_2, \ldots, N_n\}$ represent the spirals and waves drawn by the elderly in the nursing home and the lip and tongue image data. $\{Y_1, Y_2, \ldots, Y_n\}$ represent the image data of young, healthy subjects drawn by spirals, waves, lips, and tongues.

In Algorithm 1, we first use mixed sample data to build and train a sample generation model based on WGAN. Then, a sample feature preprocessing model is built. Secondly, a sample classification model is constructed and trained based on transfer learning and EfficientNetV2. Finally, the obtained Diplin model is further verified and tested using the sub-dataset wave dataset in the "Parkinson's Drawings" dataset and the "Oral Cancer (Lips and Tongue) images" dataset.

*3.5. Evaluation Metrics*

This paper uses indicators such as Accuracy, Precision, Recall, and Specificity to evaluate the Diplin model.

$$\text{Accuracy} = \frac{\text{TP} + \text{TN}}{\text{TP} + \text{TN} + \text{FP} + \text{FN}} \times 100\% \tag{5}$$

$$\text{Precision} = \frac{\text{TP}}{\text{TP} + \text{FP}} \times 100\% \tag{6}$$

$$\text{Recall} = \frac{\text{TP}}{\text{TP} + \text{FN}} \times 100\% \tag{7}$$

$$\text{Specificity} = \frac{TN}{TP + FN} \times 100\% \qquad (8)$$

$$\text{F1} = \frac{2TP}{2TP + FP + FN} \times 100\% \qquad (9)$$

In the above formulas, TP means true positive, TN means true negative, FP means false positive, and FN means false negative. The confusion matrix of the evaluation indicators is shown in Table 5.

**Table 5.** Confusion matrix.

| Sample Type | Predicted as a Normal Sample | Predicted as an Attack Sample |
|---|---|---|
| normal sample | TN | FP |
| attack sample | FN | TP |

The ROC (receiver operating characteristic) curve was first used in radar signal detection to distinguish signal from noise. Later, researchers applied it to the evaluation index to evaluate classification models' predictive ability. The ROC curve is proposed based on the confusion matrix. The ROC curve is drawn using the false positive rate (FPR) as the abscissa and the valid rate (TPR) as the ordinate.

AUC (area under ROC curve) is the area under the ROC curve. When comparing the effects of classification models, draw the ROC curve of each model and compare the area under the curve as an indicator for judging the quality of the classification model. The maximum value for this area is 1. The closer the AUC is to 1.0, the more realistic the classification model is. When the AUC value of a binary classification model is equal to or less than 0.5, the model has no authenticity and no practical application value.

This section proposes the Diplin model using Wasserstein GAN, transfer learning, and EfficientNetV2. We describe the model framework, sample data, model implementation process, algorithms, and evaluation indicators.

## 4. Experimental Results and Analysis

In this section, we describe the experiments of the proposed Diplin model based on WGAN, transfer learning, and EfficientNetV2.

### 4.1. Comparison of Transfer Learning Experiment Results

We first used Kaggle's public datasets "Parkinson's Drawings" and "Oral Cancer (Lips and Tongue) images" to compare the experimental results with and without transfer learning. The specific content is shown in Figure 4.

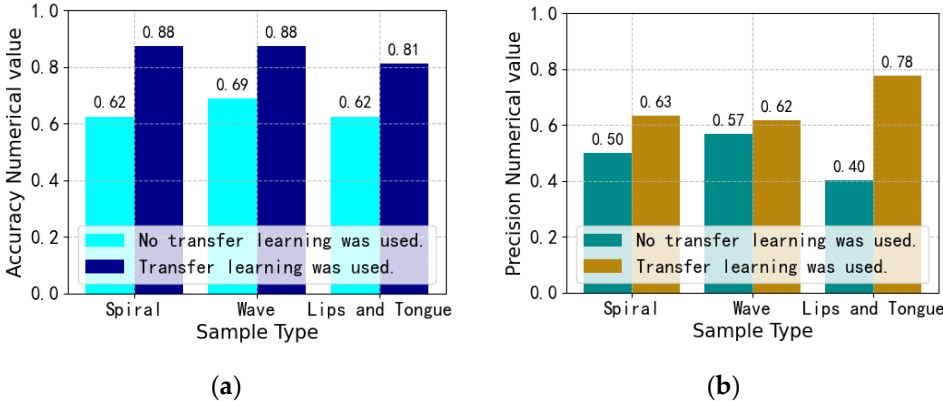

**Figure 4.** Comparison of experimental results using transfer learning and not using transfer learning. (**a**) Accuracy rate comparison. (**b**) Precision rate comparison.

Figure 4 compares the experimental results trained when the batch_size is set to 8 and learning_rate is set to 0.0001, and the optimizer uses Adam. As can be seen from Figure 4, in the case of the same dataset and the same training parameters, the accuracy and precision of using transfer learning are higher than those without using transfer learning.

### 4.2. Comparison of Experimental Results of Different Optimizers

In order to understand the impact of different optimizers on the model, in this paper, we used the spiral sub-dataset in the dataset "Parkinson's Drawings", publicly available on Kaggle, to compare the experimental results of the model using different optimizers. The specific content is shown in Figure 5.

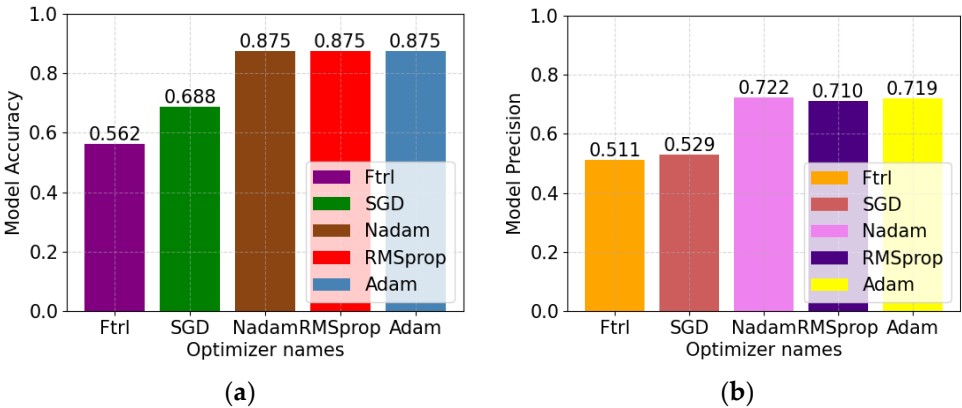

**Figure 5.** Comparison of experimental results of the model using different optimizers. (**a**) Accuracy rate comparison. (**b**) Precision rate comparison.

Figure 5 compares the experimental results trained when the batch_size is set to 8 and the learning_rate is set to 0.0001. As shown in Figure 5, the accuracy of the model using the two optimizers Nadam and Adam is relatively high, and the accuracy of the model using the two optimizers RMSprop and Adam is relatively high.

In this paper, to gain a deeper understanding of the experimental results of the model using different optimizers, we use the spiral sub-dataset in the Kaggle public dataset "Parkinson's Drawings" and compare the accuracy curve and loss curve of the model using different optimizers. The specific content is shown in Figure 6.

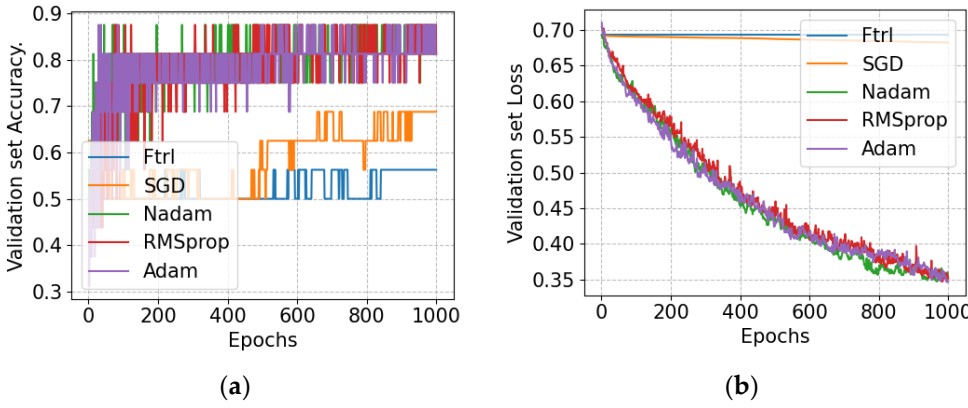

**Figure 6.** Comparison of experimental results curves of the model using different optimizers. (**a**) Accuracy curve. (**b**) Loss curve.

Figure 6 compares the experimental result curves trained when batch_size is set to 8, epochs are set to 1000, and learning_rate is set to 0.0001. Figure 6a shows that the accuracy fluctuation of the model using the two optimizers, Nadam and Adam, is relatively small.

Figure 6b shows that underfitting occurs when the model uses the two optimizers, Ftrl and SGD.

As seen from Figure 6b, the loss curve of the model using the Adam optimizer continues to decline. In order to further observe the accuracy curve and loss curve of the optimizer using Adam, we set the batch_size to 8, the learning_rate to 0.0001, and the epochs to 1300 when the optimizer uses Adam. The accuracy and loss comparison curves of the training set and the verification set are shown in Figure 7.

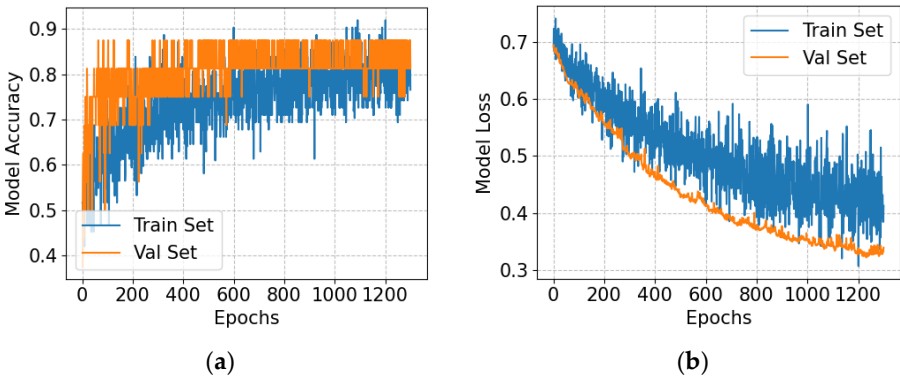

(**a**)  (**b**)

**Figure 7.** Comparison of experimental results curves for the model using the Adam optimizer on the training and validation sets. (**a**) Accuracy curve. (**b**) Loss curve.

As seen in Figure 7a, the training and validation accuracy fluctuation ranges have begun to stabilize. It can be seen from Figure 7b that both training loss and validation loss have begun to converge, and the difference between them tends to be smaller and smaller.

### 4.3. Comparison of Experimental Results with Different Learning Rates

In this paper, we used the spiral sub-dataset in the Kaggle public dataset "Parkinson's Drawings" to compare the experimental results of the model using different learning rates. The specific content is shown in Figure 8.

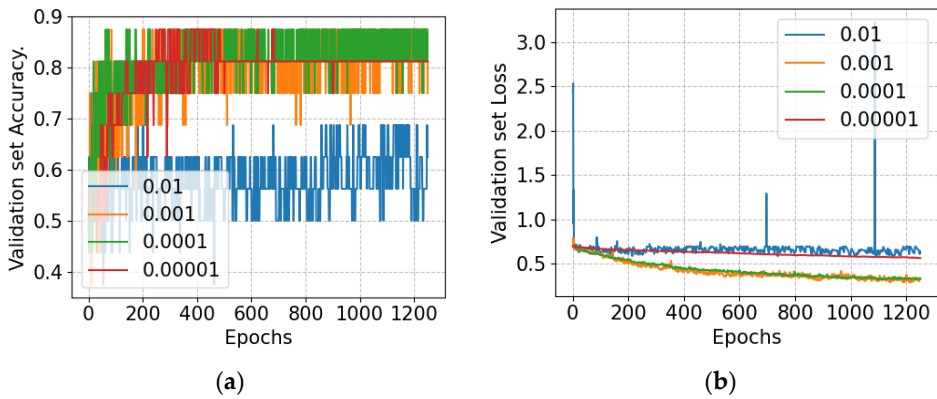

(**a**)  (**b**)

**Figure 8.** Comparison of experimental results curves for models using different learning rates. (**a**) Accuracy curves. (**b**) Loss curve.

Figure 8 compares the experimental results trained when the batch_size is set to 8, the epochs are set to 1250, and the learning_rate is set to 0.01, 0.001, 0.0001, and 0.00001, respectively. As seen from Figure 8a, when the learning_rate is set to 0.001, 0.0001, and 0.00001, the model's accuracy on the validation set is relatively high. As shown in Figure 8b, when the learning_rate is set to 0.001 and 0.0001, the loss begins to converge.

### 4.4. Model Optimization

In order to further improve the accuracy and AUC (area under ROC curve) value of the Diplin model, we optimized the composition of the sample feature preprocessing module

and the composition of the sample classification module in the Diplin model. On the spiral sub-dataset in the dataset "Parkinson's Drawings" released by Kaggle, the accuracy and AUC values of the Diplin model before and after optimization are compared, as shown in Figure 9.

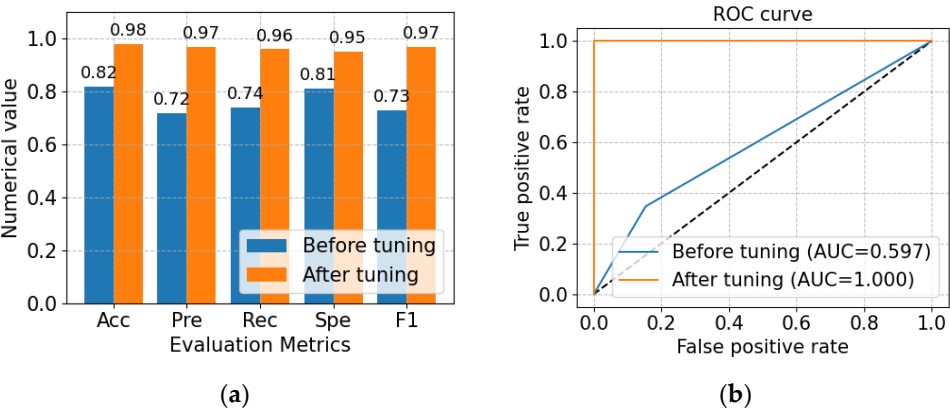

(**a**)       (**b**)

**Figure 9.** Comparison of the accuracy, precision, recall, specificity, F1 score, and AUC value of the Diplin model before and after optimization. (**a**) Comparison of accuracy, precision, recall, specificity, and F1 scores. (**b**) Comparison of AUC values.

Figure 9 compares the experimental results trained when the batch_size is set to 8 and the learning_rate is set to 0.0001. As seen in Figure 9a, the optimized model has an accuracy rate of 98%, a precision rate of 97%, a recall rate of 96%, and a specificity of 95% on the spiral dataset. The F1 score reached 97%. Figure 9b shows that the AUC value of the optimized Diplin model has reached 1.0 on the spiral dataset.

### 4.5. Comparison of Experimental Results of Different Algorithms

Standard algorithms in image classification include MobileNet, EfficientNet, Xception, DenseNet, and Inception. In this paper, to understand the practical effect of using different algorithms to build the Diplin model, we use the spiral sub-dataset in the Kaggle public dataset "Parkinson's Drawings". We built a classification model using MobileNetV3-Small, EfficientNetV2, Xception, DenseNet201, and InceptionResNetV2. In the process of building the Diplin model using the five algorithms, MobileNetV3-Small, EfficientNetV2, Xception, DenseNet201, and InceptionResNetV2, the optimizer uses Adam, the batch_size setting is 8, and the learning_rate setting is 0.0001. We compared the accuracy, precision, recall, specificity, F1 score, and ROC curve of the Diplin model constructed using different algorithms. The specific content is shown in Figures 10–12.

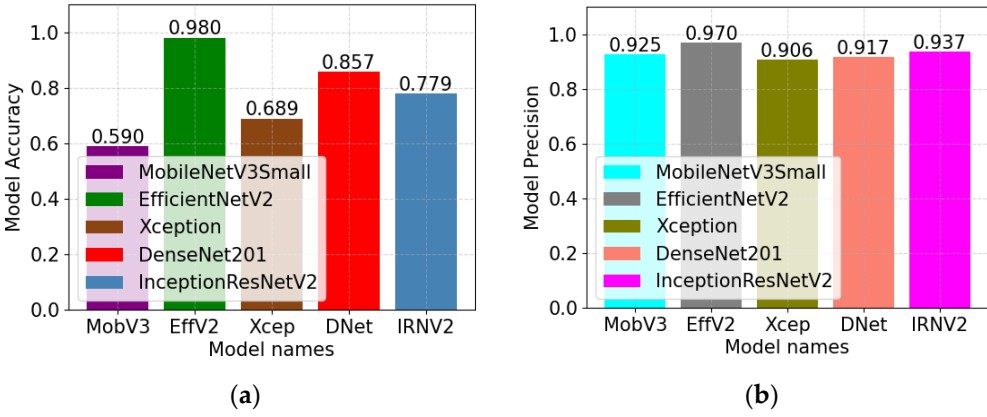

(**a**)       (**b**)

**Figure 10.** Comparison of the accuracy and precision of models built using different algorithms. (**a**) Comparison of accuracy rates. (**b**) Comparison of precision rates.

As shown in Figure 10a, the accuracy of the model built using EfficientNetV2 is the highest, reaching 98%. As shown in Figure 10b, the accuracy of the model built using EfficientNetV2 is the highest, reaching 97%.

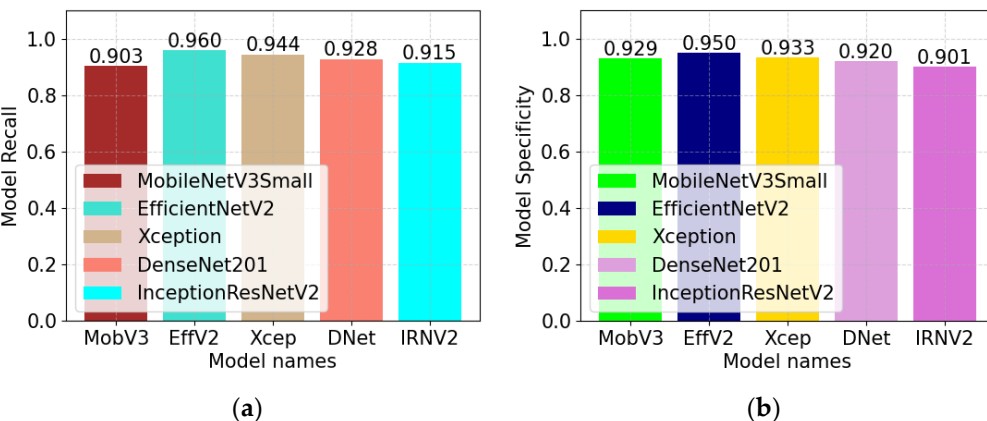

**Figure 11.** Comparison of recall and specificity of models built using different algorithms. (**a**) Comparison of recall rates. (**b**) Comparison of specificity rates.

As shown in Figure 11a, the recall rate of Diplin based on EfficientNetV2 is the highest, reaching 96%. As shown in Figure 11b, the specificity of the model built using EfficientNetV2 is the highest, reaching 95%.

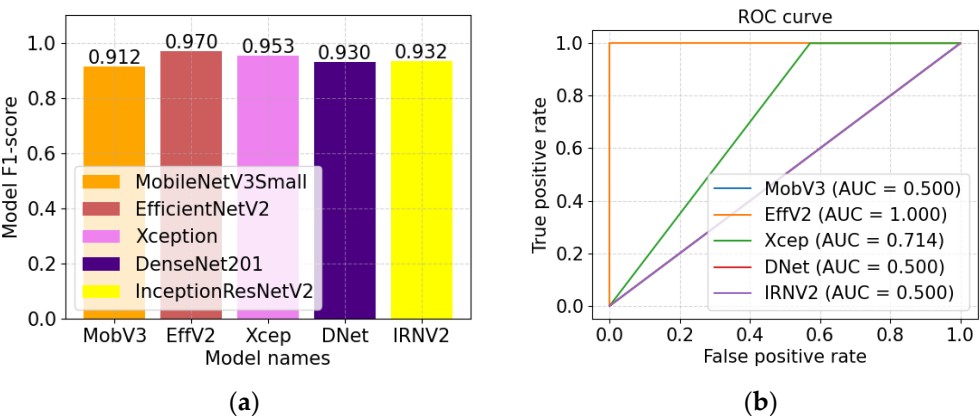

**Figure 12.** Comparison of F1 scores and ROC curves of models built using different algorithms. (**a**) Comparison of F1 scores. (**b**) Comparison of ROC curves.

As seen in Figure 12a, the F1 score of the model built using EfficientNetV2 is the highest, reaching 97%. As shown in Figure 12b, the AUC value of the model built using EfficientNetV2 is the highest, reaching 1.0.

In summary, among the five standard image classification algorithms (MobileNetV3-Small, EfficientNetV2, Xception, DenseNet201, and InceptionResNetV2), the accuracy, precision, recall, specificity, F1 score, and AUC obtained by the EfficientNetV2 algorithm in the experiment have the highest value.

### 4.6. Experimental Results on Other Datasets

According to our research, collecting images of hand-drawn curves and mouths in nursing homes is relatively convenient. Therefore, in order to further verify the reliability and accuracy of the Diplin model method, we use the wave sub-dataset in "Parkinson's Drawings" and the "Oral Cancer (Lips and Tongue) images" dataset for further verification tests. The specific content is shown in Figure 13.

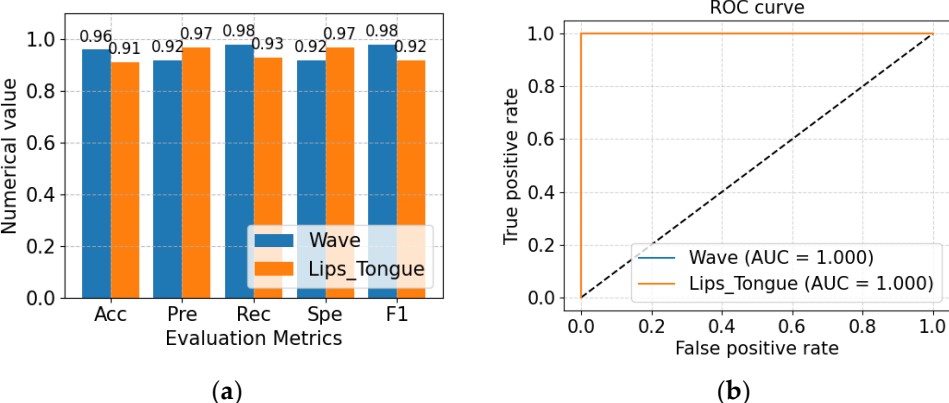

**Figure 13.** Comparison of experimental results with other datasets. (**a**) Comparison of accuracy, precision, recall, specificity, and F1 scores. (**b**) Comparison of AUC values.

Figure 13 compares the experimental results trained using the wave dataset and the "Oral Cancer (Lips and Tongue) images" dataset when the batch_size is set to 8 and the learning_rate is set to 0.0001. Figure 13a shows that the accuracy, precision, recall, specificity, and F1 score of the Diplin model method on the wave dataset and the "Oral Cancer (Lips and Tongue) images" dataset have reached more than 90%. Figure 13b shows that the AUC value of the Diplin model method on both the wave dataset and the "Oral Cancer (Lips and Tongue) images" dataset reached 1.0.

The contribution of this paper is to apply the algorithm to the actual scene in combination with the hardware configuration of the nursing home.

In this section, we conduct comparison experiments from various perspectives, such as transfer learning, optimizer, learning rate, and base algorithm. Through optimization, we obtained the best model with an accuracy rate of 98%, precision rate of 97%, recall rate of 96%, specificity of 95%, F1 score of 97%, and AUC value of 1.0. In order to further verify the method of Diplin model design and implementation, we use the wave dataset and the "Oral Cancer (Lips and Tongue) images" dataset to verify the method of Diplin model design and implementation. The above experimental data show that the Diplin model can evaluate health risks without relying on professional medical testing equipment.

## 5. Conclusions

In this paper, we use Kaggle's public datasets "Parkinson's Drawings" and "Oral Cancer (Lips and Tongue) images" as sample data and propose a Diplin model based on WGAN, migration learning, and EfficientNetV2. After optimizing the sample feature extraction module and sample classification module, the accuracy rate of the Diplin model on the verification set reached 98%, the precision rate reached 97%, the recall rate reached 96%, and the specificity reached 95%. The F1 score reached 97% and the AUC value reached 1.0. Most current research studies on disease risk prediction use datasets collected by professional medical equipment. Disease data collected by non-medical devices are minimal, and the samples used in this paper are only for Parkinson's disease and oral cancer. In the future, we hope to add more disease samples to improve the comprehensiveness of the model. In addition, the proportion of subject data is relatively low in this study. In the following research, we will further increase the subject data to improve the prediction effect of the Diplin model.

**Author Contributions:** F.Z. and S.H., wrote the main manuscript text, and Z.L. provided the idea. X.W. and J.W. prepared the data and figures. All authors reviewed the manuscript. The authors read and approved the final manuscript. All authors have read and agreed to the published version of the manuscript.

**Funding:** The work of this paper is supported by the National Key Research and Development Program of China (2019YFB1405000) and the National Natural Science Foundation of China under grant no. 61873309, 92046024, 92146002.

**Institutional Review Board Statement:** Not applicable.

**Informed Consent Statement:** Not applicable.

**Data Availability Statement:** The "Parkinson's Drawings" dataset is from https://www.kaggle.com/datasets/kmader/parkinsons-drawings, accessed on 6 March 2023. The "Oral Cancer (Lips and Tongue) images" dataset is from https://www.kaggle.com/datasets/shivam17299/oral-cancer-lips-and-tongue-images, accessed on 6 March 2023.

**Acknowledgments:** Thanks to Xiong Yu, Chief Physician of the Department of Cardiology, and Cheng Hui, Deputy Chief Physician of the Department of Orthopedics, for their medical guidance and co-operation.

**Conflicts of Interest:** The authors declare no conflict of interest.

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
