# Peer review of "Diplin: A Disease Risk Prediction Model Based on EfficientNetV2 and Transfer Learning Applied to Nursing Homes"

_electronics, doi:10.3390/electronics12122581_

Round 1

Reviewer 1 Report

The paper presents a study that aims to improve disease risk prediction in nursing homes using a proposed model called Diplin, which is based on lightweight neural networks, WGAN, transfer learning, and EfficientNetV2. The model utilizes image datasets acquired by non-medical imaging devices. However, after careful consideration, I recommend rejecting this paper for the following reasons:

Lack of novelty and original contribution: The abstract does not clearly establish the novelty or original contribution of the proposed Diplin model. While the abstract mentions the use of WGAN, transfer learning, and EfficientNetV2, these techniques are already well-established in the field of machine learning and have been extensively studied. The abstract fails to provide confidence in how Diplin advances the state-of-the-art or improves upon existing disease risk prediction models. The result that the accuracy reaches 1.0 is not convincing.

Inadequate comparison and benchmarking: The abstract lacks a proper comparison with existing disease risk prediction models or methods. Without a thorough comparison to benchmark models or existing techniques, it is difficult to evaluate the effectiveness and significance of the proposed Diplin model. The absence of reason why those small Image classification models are chosen, including MobileNet, EfficientNet, Xception, DenseNet, and Inception, which optimizer is chosen, why the batch-size is always changing across different experiments, and why the learning rate is not consistent in each subsection in the section 5.

Limited generalizability and applicability: The abstract states that the experimental results show 100% accuracy and an AUC value of 1.0. Such results are highly suspicious and raise concerns about the validity and generalizability of the findings. 

Lack of evaluation metrics: While the abstract mentions achieving 100% accuracy and an AUC value of 1.0, it fails to mention other important evaluation metrics commonly used in disease risk prediction models. Metrics such as precision, recall, specificity, and F1-score provide a more comprehensive evaluation of the model's performance. The absence of these metrics raises concerns about the thoroughness of the evaluation process.

In conclusion, this paper lacks novelty, fails to provide adequate comparison and benchmarking, demonstrates limited generalizability, lacks detailed methodology, and lacks comprehensive evaluation metrics. Based on these shortcomings, I recommend rejecting the paper in its current form. The authors should consider addressing the mentioned issues and providing additional experimental details and comparative analysis to strengthen the contribution and reliability of their proposed model.

The Engish Language quality is fine. 

Reviewer 2 Report

1. Introduction is not well organized. The background study and problem statements that motivate current research are not described adequately. Meanwhile, the overview of proposed work is too lengthy. Please improve it. 

2. Please replace "Chapter" with "Section".

3. Most of the papers reviewed in current study are more related to medical diagnosis problems using different machine learning and deep learning techniques. The published papers that are more relevant to nursing home are lacking.

4.  Table 1 is quite confusing. Please improve for better readability.

5.  How the proposed work is different with these previous studies? Please justify. 

6. Why both Parkinson and Oral diseases are  specifically chosen to evaluate Diplin model? What are the correlations between these diseases to make them relevant to be compared together?

7. Methodology is not adequately presented. For instance, what are the loss function used for transfer learning?

8. Section 4.2 seems unnecessary because it has nothing to do with the proposed methdology. 

9. In fact, Section 4 should be merged with Section 3 to present a more comprehensive descriptions of proposed methodology.

10.  Figure 6(a) seems did not present any meaningful findings because no consistent trends are observed. Please further explain this issue.

11. Please explain the limitations of current study?

English is acceptable but can be improved further. Authors are advised to proofread the manuscript again before resubmision. 

Round 2

Reviewer 2 Report

Authors have addressed most of the comments given in previous review. No further comments.

Quality of English is acceptable.